# 3-O-Methyltolcapone and Its Lipophilic Analogues Are Potent Inhibitors of Transthyretin Amyloidogenesis with High Permeability and Low Toxicity

**DOI:** 10.3390/ijms25010479

**Published:** 2023-12-29

**Authors:** Thanalai Poonsiri, Davide Dell’Accantera, Valentina Loconte, Alessandro Casnati, Laura Cervoni, Alessandro Arcovito, Stefano Benini, Alberto Ferrari, Marco Cipolloni, Elisa Cacioni, Francesca De Franco, Nicola Giacchè, Serena Rinaldo, Claudia Folli, Francesco Sansone, Rodolfo Berni, Michele Cianci

**Affiliations:** 1Bioorganic Chemistry and Bio-Crystallography Laboratory (B_2_Cl), Faculty of Agricultural, Environmental and Food Sciences, Free University of Bolzano, 39100 Bolzano, Italy; thanalai.pon@mahidol.ac.th (T.P.); stefano.benini@unibz.it (S.B.); 2Department of Chemistry, Life Sciences and Environmental Sustainability, University of Parma, Parco Area delle Scienze 17/a, 43124 Parma, Italy; davide.dellaccantera@unipr.it (D.D.); alessandro.casnati@unipr.it (A.C.); francesco.sansone@unipr.it (F.S.); rodolfo.berni@unipr.it (R.B.); 3Department of Anatomy, University of California San Francisco, San Francisco, CA 94143, USA; valentina.loconte@ucsf.edu; 4Lawrence Berkeley National Laboratory, Molecular Biophysics and Integrated Bioimaging Division, Berkeley, CA 94720, USA; 5Department of Biochemical Sciences, University of Rome “La Sapienza”, P.le Aldo Moro 5, 00185 Rome, Italy; laura.cervoni@uniroma1.it (L.C.); serena.rinaldo@uniroma1.it (S.R.); 6Department of Biotechnological Sciences and Intensive Care, Catholic University of Sacred Heart, Largo F. Vito 1, 00168 Rome, Italy; alessandro.arcovito@unicatt.it; 7Fondazione Policlinico Universitario A. Gemelli—IRCCS, 00168 Rome, Italy; 8Department of Food and Drug, University of Parma, 43124 Parma, Italy; ferro396@gmail.com (A.F.); claudia.folli@unipr.it (C.F.); 9TES Pharma S.r.l., Via P. Togliatti 20, Corciano, 06073 Perugia, Italy; mcipolloni@tespharma.com (M.C.); ecacioni@tespharma.com (E.C.); fdefranco@tespharma.com (F.D.F.); ngiacche@tespharma.com (N.G.); 10Department of Agricultural, Food and Environmental Sciences, Università Politecnica delle Marche, Via Brecce Bianche, 60131 Ancona, Italy

**Keywords:** transthyretin, amyloidogenesis inhibitors, neuronal and hepatic safety, tolcapone analogues, molecular docking and structure

## Abstract

Transthyretin (TTR) is an amyloidogenic homotetramer involved in the transport of thyroxine in blood and cerebrospinal fluid. To date, more than 130 TTR point mutations are known to destabilise the TTR tetramer, leading to its extracellular pathological aggregation accumulating in several organs, such as heart, peripheral and autonomic nerves, and leptomeninges. Tolcapone is an FDA-approved drug for Parkinson’s disease that has been repurposed as a TTR stabiliser. We characterised 3-O-methyltolcapone and two newly synthesized lipophilic analogues, which are expected to be protected from the metabolic glucuronidation that is responsible for the lability of tolcapone in the organism. Immunoblotting assays indicated the high degree of TTR stabilisation, coupled with binding selectivity towards TTR in diluted plasma of 3-O-methyltolcapone and its lipophilic analogues. Furthermore, in vitro toxicity data showed their several-fold improved neuronal and hepatic safety compared to tolcapone. Calorimetric and structural data showed that both T4 binding sites of TTR are occupied by 3-O-methyltolcapone and its lipophilic analogs, consistent with an effective TTR tetramer stabilisation. Moreover, in vitro permeability studies showed that the three compounds can effectively cross the blood-brain barrier, which is a prerequisite for the inhibition of TTR amyloidogenesis in the cerebrospinal fluid. Our data demonstrate the relevance of 3-O-methyltolcapone and its lipophilic analogs as potent inhibitors of TTR amyloidogenesis.

## 1. Introduction

Transthyretin (TTR) is a 55 kDa amyloidogenic homotetramer involved in the transport of thyroxine (T4) in the blood and cerebrospinal fluid [1,2]. Wild-type (WT) TTR has an inherent amyloidogenic potential [3], leading to Senile Systemic Amyloidosis (SSA), an under-diagnosed disease that manifests itself in the elderly and is mainly characterised by cardiomyopathy [4]. Both WT-TTR and its more aggressive mutant forms give rise to TTR-related amyloidosis (ATTR). Mutations are responsible for hereditary ATTR, whose phenotypes may affect different organs, such as heart, peripheral and autonomic nerves, and leptomeninges [5]. The generation of amyloid fibrils is the result of the rate-limiting dissociation of the TTR tetramer, followed by the misfolding of TTR monomers and subsequent aggregation [6,7,8,9].

To date, only a few strategies have been adopted for the therapeutic treatment of ATTR. These includes liver transplantation, stabilisation of the TTR tetrameric native structure by specific ligands, and, more recently, therapies based on TTR-specific siRNA and anti-sense oligonucleotides. Patisiran is an RNA interference therapeutic, administered as intravenous infusion, which hinders the production of both mutant and nonmutant forms of TTR [10]. Inotersen is an antisense oligonucleotide that complements exactly the messenger RNA (mRNA) that encodes for TTR [11].

TTR stabilisers establish interactions by bridging the two TTR monomers that form each T4 binding site present in the oligomeric protein, thereby stabilising its tetrameric native state and inhibiting amyloidogenesis [6,12,13,14,15,16,17,18]. 

Several natural compounds are listed as inhibitors of transthyretin amyloidosis and neuroprotective agents [19], but, currently, there are only few TTR ligands approved or under investigation for the ATTR pharmacological therapy (Figure 1A). Tafamidis is an approved drug that can delay the disease progression in patients at the early stages of ATTR polyneuropathy [20,21] and cardiomyopathy [22]. Diflunisal, a nonsteroidal anti-inflammatory drug, has been documented as effective for halting ATTR amyloidosis progression by stabilising the tetramer form of TTR. The use of these two drugs led to improvements in polyneuropathy and cardiac failure [23,24,25]. AG10 and tolcapone are among the new promising alternatives. 

AG10 is undergoing clinical trials, and it has been developed for the treatment of TTR amyloidosis cardiomyopathy (ATTR-CM) by stabilising WT-TTR and the V122I-TTR mutant, which are related to the development of familial amyloid cardiomyopathy [13]. It has been shown that AG10 mimics the protective effect of the non-pathogenic TTR variants T119M [26] and R104H [27]. AG10 stabilises the heterotetramers of these variants, WT-TTR, and TTR amyloidogenic mutants, preventing in vitro amyloidogenesis. 

Tolcapone (Figure 1A), an FDA-approved drug for Parkinson’s disease, has been repurposed as a potent and selective TTR stabiliser [15]. Tolcapone possesses an ex vivo anti-amyloidogenic activity higher than that of tafamidis, making it potentially effective for treating TTR amyloidosis.

Due to the ability of crossing the blood-brain barrier [28], tolcapone inhibits the aggregation of the V30M TTR variant that cause CNS amyloidosis [29,30]. However, while the V30M TTR variant is responsible for earlier onset of CNS amyloidosis, its amyloid deposits consist mainly of full-length WT-TTR [31,32]. These findings are consistent with the notion that the high amyloidogenic potential of human pathogenic TTR variants, especially when mutations are located in β-strands, is determined by the destabilization of their native structures, while WT-TTR alone has a high intrinsic β-aggregation propensity, which is not enhanced by amyloidogenic mutations [3]. As such, while stabilization of TTR variants might delay the onset of amyloid aggregation, stabilization of WT-TTR delays its progress towards amyloidogenesis [33].

In this context, tolcapone is also hepatotoxic [34,35,36] and has a short half-life in the plasma (about 3 h), which can be ascribed to rapid glucoronidation of the free 3-hydroxy group and its consequent rapid elimination via the urinary tract [37]. Only the 3-hydroxy group of tolcapone is subject to glucorinidation since no glucorinidation products for the free 4-hydroxy group have been observed [37]. The hepatotoxicity of tolcapone has also been linked to the glucuronidation activity of the enzyme UDP-glucuronosyl transferase [38]. On the other hand, 3-O-methyltolcapone (**3-OMT**), a metabolite of tolcapone, has remarkably more favourable pharmacokinetics in humans than tolcapone [37].

In 2020, we reported the synthesis of **3-OMT** and 3-deoxytolcapone as potential TTR stabilizers, establishing structure-activity relationships for both molecules, for which the lack of a free 3-hydroxy group would impede glucuronidation [39]. On the basis of a Western Blot analysis, in the presence of plasma proteins, **3-OMT** were more effective and selective in comparison to 3-deoxytolcapone at stabilizing the tetrameric native structure of human plasma TTR, and with comparable results to those of tolcapone, in agreement with their very similar chemical structures and the limited differences in their interactions with TTR [39]. The methylation of the free 3-hydroxy group in tolcapone should also confer greater lipophilicity in comparison to tolcapone, thus potentially increasing brain permeability for the pharmacological therapy of CNS TTR amyloidosis (leptomeningeal and oculo-leptomeningeal amyloidosis). 

In the present work, starting from the previously reported structure-activity relationships of **3-OMT** in complex with TTR [39], we proceeded to design five **3-OMT** analogues (Figure 1B). While recently reported new tolcapone analogues conserve its 3-4-dihydroxy-5-nitrophenyl ring but present modifications in the non-phenolic aromatic ring to achieve the highest affinities [18], our analogues maintain the 4-hydroxy-3-methoxy-5-nitrophenyl of **3-OMT** to avoid glucuronidation and focus on the non-phenolic aromatic ring of tolcapone to achieve lower toxicity and greater permeability and lipophilicity without worsening the affinity. This approach would generate tolcapone-like candidates for therapeutic intervention of both familial amyloid polyneuropathy and CNS amyloidoses. Based on the results of docking studies, the best candidates, were synthetized and tested for in vitro and ex vivo TTR stabilisation, permeability, and cytotoxicity. Finally, the crystallographic analysis of the TTR-ligand complexes rationalised the key interactions between ligand and TTR and, consequently, the molecular basis of the effective TTR stabilisation in vitro.

Among all the TTR stabilizers reported in the literature, these are among the first to be designed following the principle of avoiding glucuronidation.

## 2. Results and Discussion

### 2.1. Design of 3-OMT Lipophilic Analogues

The native tetrameric structure of TTR is a dimer of homodimers related by a 222 symmetry. Each pair of twofold-related monomers forms the hormone-binding site, a central cavity at the dimer-dimer interface, crossing the entire complex, which is lined with two sets of three hydrophobic halogen binding pockets (HBPs) where the iodine atoms of the bound T4 molecules are harboured [2,40]. The three binding pockets are distinguished into an outer binding subsite (HBP1 and HBP1′, with 1 and 1′ indicating the two constitutive dimers), an inner binding subsite (HBP3 and HBP3′), and an intervening interface (HBP2 and HBP2′). HBP1 is more hydrophilic and is composed of the residues Lys15, Leu17, Thr106, and Val121. On the other hand, HBP2 is mostly hydrophobic and is formed by Leu17, Ala108, Ala109, and Leu110 along with the side chain of Lys15. Finally, HBP3 is in the innermost area of the binding pockets and is formed by the side chains of Ser117, Leu110, Thr119, and Ala108.

Both tolcapone [15] and **3-OMT** [39] bind TTR in the so-called “forward binding mode”. In the case of **3-OMT**, this pose is characterised by the position of the polar moiety of the 3-methoxy-4-hydroxy-5-nitrophenyl ring in HBP1 and HBP1′, forming a hydrogen bond with Lys15, which in turn stabilises the favourable ionic interactions between Lys15 and Glu54. The 4-methyl-phenyl ring in both molecules is instead positioned deeply into the TTR hydrophobic HBP2/HBP2′ and HBP3/HBP3′ pairs, sandwiched between the residues Leu110 and Thr119. Given the conserved poses observed for both tolcapone and its analogue, we designed five molecules (Figure 1B) characterised by the identical polar 3-methoxy-4-hydroxy-5-nitrophenyl ring and by a variable number (two or three) of methyl groups in different positions in the lipophilic phenyl unit, with the aim of increasing compound lipophilicity, permeability through the blood-brain barrier, binding selectivity, and structural stabilisation. Nonpolar groups (e.g., methyl) are typically added to a molecule to enhance lipophilicity [41]. LogP values determination for tolcapone and its analogues confirms that lipophilicity is increasing (Appendix A) as far as the number of methyl groups increases from 3-OMT to compounds **1**–**3** and to **4** and **5** (one, two, and three methyl groups, respectively).

### 2.2. In Silico Docking of 3-OMT Lipophilic Analogues

The interaction of TTR protein structure with the designed analogues was first validated with in silico docking analysis of the five separate compounds (Figure 1B). The 3D structure of TTR tetramer is well-known and the amino acids exposed in the binding pockets undergo only local rearrangements during binding process. Knowing the binding pocket and the amino acids originally exposed to the interaction with **3-OMT** [39], we used in silico docking [42] to anticipate the binding mode of the ligand, as previously performed for TTR [43,44,45]. This is a less computationally demanding alternative to Molecular Dynamics, used recently to evaluate the pose of ligands within this protein [18]. We verified that the simulation was able to reproduce the original ligand binding using the **3-OMT** as a testbed. The top-scoring structures of the five compounds were derived from a limited search in the known protein halogen binding pockets [39,46]. The top-scoring binding molecules were identified by fulfilling the criteria of lower ΔG_binding_ and FullFitness (FF) values (Appendix A). Although all compounds showed higher hydrophobicity than tolcapone and **3-OMT** (Appendix A), the favoured interactions were verified for analogues **1** and **2**, which showed higher affinity for the HBPs. The binding site for these two compounds is formed by residues at the bottom of HBP3 and HBP2, involving Ser117, Leu110, Thr119, and Leu17 that interact with the hydrophobic ring of the compounds, whereas the hydrophilic ring is stabilised by Lys15. The position of the binding pocket and the type of interactions involved in the stabilisation of the compounds are consistent with those observed with tolcapone [15] **3-OMT** [39], flurbiprofen [46], and tafamidis [12]. The docking identified the most favourable binding mode as “the forward one”, in agreement with prior findings on lipophilic analogues of tolcapone [15]. Instead, analogues **3** and **4** showed lower affinity to the HBPs in terms of binding energy and FF, whereas analogue **5** did not show any affinity for the HBPs. We speculated that analogues **1** and **2** showed lower binding energy thanks to the lower steric clashes when compared to the other compounds. Similarly, analogues **4** and **5** were likely to raise the binding energy by increasing the steric hindrance. The docking poses of analogues **1** and **2** were later experimentally confirmed.

### 2.3. Synthesis of 1 and 2

Based on molecular docking results and LogP prediction, we synthesised compounds **1** and **2** (Figure 2). Following the approach of Soares-da-Silva et al. [47], we started from benzyl protected vanillin **6** (Figure 2A), which was reacted with the Grignard reagent prepared from 3,5-dimethylbromobenzene to obtain alcohol **7** after quenching with NH_4_Cl. Oppenauer–Woodward oxidation on **7** was carried out to obtain ketone **8** using sodium tert-butoxide as base and cyclohexanone as hydride acceptor in refluxing toluene. Ketone **8** was deprotected from the benzyl group by catalytic hydrogen transfer using ammonium formate as hydrogen donor and Pd/C (10%) as catalyst to give **9**. Nitration with 65% nitric acid in glacial acetic acid was carried out to produce compound **1**. No products of nitration in other positions were observed. 

The synthesis of isomer **2** through the same pathway (Appendix A) failed. During the final nitration step, we isolated only 2,4-dinitro-6-methoxyphenol instead of the expected mononitro-diphenyl-ketone. We attributed this outcome of the reaction to the different position of the methyl groups in the analogue (**13** in Appendix A) of **9** that seems to allow an ipso-nitration with the release of the stabilised 2,4-dimethyl-benzoyl cation (Appendix A). Therefore, an alternative strategy was applied (Figure 2B). Oxidation of vanillin to vanillic acid and subsequent nitration in the same conditions used for **9** furnished 5-nitrovanillic acid **10** [48]. This was transformed in the corresponding acyl chloride that was immediately reacted, without isolation, with m-xylene to provide the target molecule **2**.

### 2.4. Chromatographic Hydrophobicity Index (CHI)

Table 1 reports the CHI values for the tested compounds. At pH = 2 the molecules are present in neutral form. All three compounds (**3-OMT**, **1** and **2**) were found to be more hydrophobic (CHI > 85) than tolcapone. Having one less methyl group, **3-OMT** is slightly more polar than the other two derivatives **1** and **2**, as predicted by LogP. At physiological and basic pH, the most abundant species for the three 3-methoxytolcapones are their conjugated bases; they turn out to be rather hydrophilic, showing a CHI value around 50. The three CHI values at different pH, taken together, define that the three molecules have a rather acidic character following the K. Valko classification [49]. Tolcapone, on the other hand, showed a more complex profile due to the presence of two hydroxyl groups. At pH 7.4, the chromatographic peak extended over a large time interval and therefore the CHI value was not determinable with this method. However, considering the condition at acidic pH, where the molecules should be ideally neutral, the CHI values indicate that the intrinsic hydrophobicity of the three derivatives (**3-OMT**, **1**, and **2**) is higher than that of tolcapone. 

### 2.5. 3-OMT and Its Lipophilic Analogues Have Reduced In Vitro Neuronal- and Hepato-Toxicity Compared to Tolcapone

It has already been reported [50,51] that tolcapone showed a cytotoxic activity in neuronal-derived cell line SH-SY5Y, a widely used in vitro model to study neurotoxicity, oxidative stress, and neurodegenerative diseases. SH-SY5Y cell viability was then determined by measuring ATP levels after 48 h of stimulation with tolcapone, **3-OMT**, and its lipophilic analogues **2** and **1** at different concentrations.

In our hands, tolcapone showed cytotoxic activity in an ATP viability assay in the SH-SY5Y neuroblastoma human cell line with an EC_50_ of 29.8 µM; this in agreement with previously reported data [50,51]. The new derivatives showed an increased EC_50_ with respect to tolcapone, suggesting an improved and non-cytotoxic profile of the molecules as calculated from dose-response curves reported in Figure 3. While tolcapone compromised cell viability at low micromolar concentrations, all three 3-OMT-based derivatives demonstrated 6-7-fold reduced toxicity in SH-SY5Y cells (EC_50_ values), as reported in Table 2, indicating an increase in their safety profile. Considering that tolcapone has been described as a hepatotoxicity inducer [35], a further study to characterise the potential effects of the three 3-O-methyl derivatives in liver cells was conducted. In HepG2 cells, tolcapone showed the same toxicity as tamoxifen, used as the positive control of the assay, and the three 3-OMT-based derivatives were less cytotoxic than tolcapone, ranging from 13- to 24-fold, with compound **1** showing the lowest hepatotoxicity (Figure 3 and Table 2).

### 2.6. 3-OMT and Its Lipophilic Analogues Stabilise TTR in Human Plasma

The TTR stabilisation coupled with binding selectivity towards TTR of tolcapone, **3-OMT**, and its analogues **1** and **2** was evaluated directly in human plasma. This is a more complex environment when compared to in vitro assays conducted with purified TTR, where binding selectivity is not evaluated due to the absence of the other plasma proteins (i.e., albumins, globulins, etc.). Aliquots of diluted human plasma were supplemented with different concentrations of tolcapone, **3-OMT**, **1** and **2**. Tafamidis was used as a reference ligand. TTR stabilisation was estimated by measuring the monomer abundance after sample incubation in slightly denaturing conditions (Figure 4). At the concentration of 10 µM, all the compounds showed a strong stabilising effect, and TTR monomer was about 10% of that present in the control sample containing dimethyl sulfoxide (DMSO). At the concentration of 1 µM, larger differences among the compounds were observed. Tolcapone, and especially **3-OMT**, displayed a greater inhibitory capability (27% and 14% of TTR monomer, respectively) compared to tafamidis (47% of TTR monomer). The additional methyl group in position 2 of the phenyl ring of **2** increases the stabilising effect (7% of TTR monomer). On the other hand, a different position of the two methyl groups, as for **1**, determines a stabilising effect comparable to that of **3-OMT** (17% TTR monomer). Among all the ones tested, ligand **2** appears to be the most promising TTR stabilising compound in human plasma.

### 2.7. Binding Affinities and Thermodynamics of Interactions

The specificity and the thermodynamics of the interaction of tolcapone, **3-OMT**, tafamidis, **1**, and **2** to TTR were directly analysed using isothermal titration calorimetry (ITC). Briefly, 5 μM protein solution was titrated with 100 µM of each compound and the titration profile is depicted in Figure 5A–E. The dissociation constant (K_d_), the enthalpy, and the stoichiometry of binding, obtained by fitting the enthalpy curve according to a single-binding mode, are reported in Table 3. The stoichiometry of binding for the tested compounds was close to 2:1 ligand:TTR tetramer. As expected for specific binding, integration of the titration peaks produced a sigmoidal enthalpy curve for each interaction (Figure 5A–E). The affinity and enthalpic binding contribution measured for tolcapone were K_d_ = 26 nM and ΔH = −11.8 kcal mol^−1^. The presence of a methoxy group instead of OH in position 3 of tolcapone to give **3-OMT** results in a slight weakening of the binding, with a K_d_ = 33 nM and ΔH = −8.9 kcal mol^−1^. The presence, with respect to **3-OMT**, of an additional methyl group in ortho position of the lipophilic phenyl ring has a positive effect for **2** on both the affinity and enthalpic binding values with K_d_ = 25.0 nM and ΔH = −10.5 kcal mol^−1^. Finally, for **1**, with the two methyl groups having a different disposition compared to **2** on the phenyl ring, the affinity was lowered (K_d_ = 71 nM) while the enthalpic binding contribution (ΔH = −11.5 kcal mol^−1^) becomes more favourable.

Previous ITC studies used different models for interpolating ITC data. The calorimetric data of the binary complexes of tafamidis with TTR were interpolated using a sequential model [12,18], revealing a strong negative cooperativity between the two binding sites. The result showed that the first binding site had an extremely low K_d_′ of 9.9 nM and a ΔH = −6.0 kcal mol^−1^, while the second binding site resulted in a loss of affinity with a very high K_d_″ of 260 nM and a ΔH = −6.5 kcal mol^−1^. In our hands, when fitting the enthalpy curve according to a single-binding mode, we obtained a K_d_ of 128 nM and ΔH = −3.2 kcal mol^−1^.

Another study of the binary complexes of tolcapone [15] with TTR using a sequential model showed a weak negative cooperativity between the two binding sites, with a very low first K_d_′ of 21 nM and ΔH = −9.7 kcal mol^−1^, and a second higher K_d_″ of 58 nM and ΔH = −9.7 kcal mol^−1^. For AG10, calorimetric data were plotted and fitted using the standard single-site binding model resulting in K_d_ = 4.8 nM with ΔH = −13.6 kcal mol^−1^ [52]. Recent ITC studies on halogenated derivatives of tolcapone that interpolated the data with the standard single-site binding model showed no cooperativity with a K_d_ of 6.2 nM and ΔH = −16.6 kcal mol^−1^ [18].

Through analysis of the thermodynamic signature using ITC, it has been proposed that selectivity is more closely correlated with the enthalpic component of the interaction than with the K_d_ [52,53]. According to our findings, compound **2** exhibits the same affinity and comparable enthalpic binding values as tolcapone (Table 3), confirming the selectivity towards TTR, as shown in the immunoblotting experiments (Figure 4). Finally, tolcapone, **3-OMT**, **1**, and **2** do not show cooperativity, so they bind to both TTR T4-binding sites significantly better on average than tafamidis (Table 3), as already observed in the case of other tolcapone analogues [18].

### 2.8. Binding Interactions between 1 and WT Human TTR

The crystal structure of the TTR/**1** complex, determined at 1.19 Å resolution, reveals the so-called “forward binding mode” (Figure 6), characterised by the position of the polar moiety of the 3-methoxy-4-hydroxy-5-nitrophenyl ring in the outermost part of the binding cavity (Figure 6), as already reported for the TTR-tolcapone complex [15] and for the **3-OMT** complex [39]. The root-mean-square deviation (rmsd) of the Cα atoms between equivalent residues (116 in total) in the monomer of the TTR/**1** complex and in the tolcapone and **3-OMT** complexes is 0.319 Å and 0.175 Å, respectively. The two binding cavities were refined equally occupied (0.50/0.50) and the electron densities for both ligands define the position of the 3-methoxy-4-hydroxy-5-nitrophenyl ring and the innermost apolar 3,5-dimethyl-phenyl ring. The 3,5-dimethyl-phenyl ring of **1** is deeply embedded in the inner hydrophobic binding sites HBP3, sandwiched between Leu110 and the γ-methyl group of Thr119′ side chain on one side and Leu110′ and the γ-methyl group of Thr119 side chain on the other side (Figure 6), with the 3,5-dimethyl groups pointing towards Ser117/Ser117′. The central carbonyl group of **1** is oriented towards the γ-hydroxyl group (3.4 Å) of Thr119′ in the HBP2′. Similar to the TTR-tolcapone complex, the 3-methoxy-4-hydroxy-5-nitrophenyl ring remains in HBP1, held in place by hydrophilic interactions with Lys15/Lys15′ and Thr119. The 3-methoxy group is 3.17 Å from the γ-amino group of Lys15 and 2.90 Å from the γ-amino group of Lys15′. The 4-hydroxy moiety is 2.75 Å from the γ-amino group of Lys15′. The 5-nitrophenyl group is tethered by the γ-hydroxyl group of Thr119. The positive charges of Lys15/Lys15′ are counterbalanced by carboxylic groups of Glu54/Glu54′ (3.01 Å).

### 2.9. Binding Interactions between 2 and WT Human TTR

The crystal structure of the TTR/**2** complex, determined at 1.10 Å resolution, confirms, similar to **1**, the “forward binding mode” (Figure 6). The Cα rmsd between equivalent atoms (116 residues) in the monomer of TTR/**2** complex and in the tolcapone and **3-OMT** complexes is 0.75 Å and 0.71 Å, respectively. The 2-fold axis running through the binding pocket generates two symmetrical binding modes of the ligand, which are rotated by 180 degrees in relation to one another. The F_o_–F_c_ electron density map at 3 σ level clearly defines both the 3-methoxy-4-hydroxy-5-nitrophenyl ring and the innermost apolar 2,4-dimethyl-phenyl ring, and the two binding cavities are equally occupied (0.50/0.50). The 2,4-dimethyl-phenyl ring of **2** is positioned deeply in the inner hydrophobic binding sites HBP3, sitting between Leu110 and the γ-methyl group of Thr119′ side chain on one side and Leu110′ and the γ-methyl group of Thr119 side chain on the other side (Figure 6), with the 4-methyl group of the 2,4-dimethyl groups pointing towards Ser117/Ser117′ and the 2-methyl group of the 2,4-dimethyl groups pointing towards Ala108′-Ala109′. The central carbonyl group of **2** is pointing towards the γ-methyl group of Thr119′ and the β-methyl group of Ala108′ in the HBP2′. The 3-methoxy-4-hydroxy-5-nitrophenyl ring remains in HBP1 and is held in place by hydrophilic interactions with Lys15/Lys15′. The oxygen of the 3-methoxy group is at 3.17 Å distance from the γ-amino group of Lys15 and the methyl group is pointing toward the γ-methyl group of Ala121′ and the γ-methyl group of Ala106′. The 4-hydroxy moiety is at 2.77 Å from the γ-amino group of Lys15′ and at 2.9 Å from the γ-amino group of Lys15. The 5-nitrophenyl group is tethered by the γ-amino group of Lys15′ at 3.0 Å away. The positive charges of Lys15/Lys15′ are counterbalanced by carboxylic groups of Glu54/Glu54′ at 2.87 Å. 

Overall, the pose of **2** is distinguished from the pose of **1** in respect to the orientation and position of the dimethyl-benzene ring, which, for the former, binds somewhat deeper into the HBP3/HBP3′ pockets (Figure 7A) and mimics the pose of tolcapone and **3-OMT** (Figure 7B).

### 2.10. In Vitro Intestinal and Blood-Brain Barrier Permeability, and Solubility

The cellular-based Caco-2 assay is a mature method to mimic and investigate human intestinal permeability and drug efflux. The rate of molecules transported across the Caco-2 cell was assessed in both directions, apical to basolateral (A→B) and basolateral to apical (B→A). All four tested compounds showed good permeability and no efflux active transport (Table 4). Tolcapone showed the lowest permeability compared to the other three compounds, with the apparent permeability P_app_ value (A→B) of 101 nm/s. **3-OMT** showed better permeability with a P_app_ value (A→B) of 160 nm/s; **1** and **2** displayed a 2.5-fold increase of permeability P_app_ value (A→B) compared to tolcapone, with a P_app_ value (A→B) of 240 nm/s and 250 nm/s, respectively. The addition of a second methyl group to **3-OMT** did not significantly affect the solubility of the compound (Table 5), which stayed above 500 μM.

The molecule ability to permeate the BBB and evade the efflux machinery, which are arranged at the apical surface of endothelial cells to shield the brain from xenobiotics, is essential for drug disposition within the CNS. The parallel artificial membrane permeability assay (PAMPA) using porcine brain lipid extracts was used to simulate BBB permeation (PAMPA-BBB). PAMPA relies on the principle of transcellular passive diffusion, which is the main mechanism for exogenous brain uptake of small molecules [54] The results of the reference compounds were in perfect agreement with the expected values. All four tested compounds showed the ability to passively cross the blood-brain barrier (Table 5). **3-OMT**, **1**, and **2** showed permeation (Pe) values that are close to the highest value obtainable with this in vitro model (~20 × 10^−6^ cm/s). **3-OMT**, **1**, and **2** showed values of permeability 2.5 times higher when compared to the permeability values measured here (Table 5) for tolcapone and those already reported [55,56]. All four compounds were highly soluble (>500 µM) in PBS at physiological pH (Table 5). While Pampa-BBB indicates excellent passive permeability, the Caco-2 experiment allowed us to exclude that active efflux transport may be involved in the decrease of BBB passage. Taking these in vitro results and the CHI data together (Table 1), we could predict optimal CNS permeability.

### 2.11. Overall Performance of 3-OMT and Its Lipophilic Analogues in Stabilising Human Transthyretin

The selectivity of TTR-stabilising ligands in human plasma can be correlated with enthalpic forces, whereby ligands with larger negative ΔH have a proportionally higher selectivity compared to ligands with a lower influence of ΔH [53]. The values of the relative abundance of TTR monomer after supplementation of ligands in plasma (Figure 4) correlates with the binding enthalpy contributions; to lower values of former for **3-OMT**, **1**, **2**, and tolcapone correspond larger negative values of the latter when compared to tafamidis (Table 3). Taken together, the values of relative abundance of TTR monomer as measured in plasma (Figure 4) and the larger negative ΔH (Table 3) support the stabilization of TTR by **3-OMT**, **1**, and **2**. 

**3-OMT** efficiently and selectively stabilised the TTR tetramer at the dimer-dimer interface to an extent equivalent to that of tolcapone because of the similar interactions of the two ligands with TTR (Figure 7B) [39]. The data presented here show that structural stabilisation of human TTR in diluted human plasma samples supplemented with **3-OMT** are better than with tolcapone (Figure 4). **3-OMT** shows a slight weakening of the binding (Table 3), with a K_d_ of 33 nM and ΔH = −8.9 kcal mol^−1^, whereas the affinity and enthalpic binding measured values for tolcapone were higher (Kd = 26 nM and ΔH = −11.8 kcal mol^−1^). 

The presence of an additional methyl group on the lipophilic phenyl ring with respect to **3-OMT** results, for compounds **1** and **2**, in lower relative abundance of TTR monomer (Figure 4) and different binding strength to TTR (Table 3). In diluted human plasma samples, **1** shows structural stabilisation of human TTR that are inferior to **3-OMT** (Figure 4), with a higher K_d_ of 71 nM compared to 33 nM for **3-OMT** (Table 3). On the other hand, **2** shows higher structural stabilisation of human TTR than **3-OMT** and **1** (Figure 4). The binding strength of **2** (Table 3) is comparable to that of tolcapone with a K_d_ of 26 nM. 

The experimentally determined poses of **1** and **2** are in agreement with the results of the starting docking studies. Both TTR complexes with compounds **1** and **2** display short Ser117/Ser117′ distances, with **2** being shorter than **1**, bridging the symmetry-related monomers by positioning the two methyl groups of **1** and **2** in the hydrophobic pockets HBP3/HBP3′ (Figure 6), thus stabilising the complex with hydrophobic interactions. In this study, the X-ray structure of TTR bound to **2** reveals distances of 4.5 Å between chain A/A′ and 5.0 Å between chain B/B′. Similarly, the TTR/**1** complex indicates the distances of 4.9 Å between chain A/A′ and 5.0 Å between chain B/B′. The shorter distance between the γ-OH of Ser117 and Ser117′, which are the opposite residues at the interface, may indicate the structure stability. In comparison, in the crystal structure of WT-TTR bound to tolcapone (PDB ID: 4D7B) [15], the distance between chain A/A′ is 5.2–5.6 Å and chain B/B′ is 5.1–5.8 Å. A greater distance was observed from WT-TTR bound to **3-OMT** structure (PDB ID: 6SUH) [39] with the value of 5.4–5.9 Å and 5.7–5.8 Å between chain A/A′ and B/B′, respectively.

When placing the relative abundance of TTR monomer after supplementation of tolcapone analogues at two different concentrations values (Figure 4) in increasing order, we obtain the sequence **2** > **1** ≈ **3-OMT** > tolcapone, which correlates well with the values of the γ-OH of Ser117/Ser117′ distances observed. Ser117/Ser117′ distances for **2**, shorter when compared to **1**, could be attributed to the fact that the TTR/**2** complex structure presents hydrogen bonds mediated by water molecules at both ligand sites, while the TTR/**1** complex structure has hydrogen bonds mediated by water molecules at just one ligand site (Figure 6). A shorter value of the γ-OH of Ser117/Ser117′ distance might also be correlated to a possible decreased freedom of motion and the release of bound waters, which are sought-aftereffects in the case of TTR-tetramer stabilisation [57]. It has been previously observed in the crystal structure of the stabilising T119M−TTR variant (PDB ID: 1FHN) [58] to have rather short distances of 4.6 Å and 5.0 Å between the γ-OH of Ser117/Ser117′ chain A and chain B, respectively [58] Even shorter values of 4.5 Å and 4.8 Å have been measured in the structure of WT-TTR bound to AG10 (PDB ID: 4HIQ) [13], attributed to the TTR tetramer stabilising properties of AG10 [52].

Analysis of the metabolism and excretion [37] clarified that tolcapone is almost completely metabolised before being excreted in urine and faeces, where it is found as 3-O-β-D-glucuronide as the most abundant metabolite. After 12 h from administration, the major metabolite of tolcapone found in plasma is **3-OMT**. The half-life reported for **3-OMT** was 40.9 h, compared to 3.3 h for tolcapone, due to the lack of the glucuronidation susceptible 3-hydroxy group in the nitro-phenyl ring [37]. Moreover, the hepatotoxicity of tolcapone has been linked to the enzyme responsible for glucuronidation, UDP-glucuronosyl transferase [38], bridging the fast excretion of tolcapone to its toxicity. Our in vitro results show that **3-OMT** and its two derivatives **1** and **2** are less toxic than tolcapone, ranging from 13- to 24-fold (Table 2). The substitution of the OH group in position 3 of tolcapone with a methoxy group to create **3-OMT** not only prolongs the half-life [37] and reduces toxicity (Table 2), but also increases three-fold the intestinal and the blood-brain barrier permeability (Table 4 and Table 5), albeit while worsening the affinity. The effect of adding an extra methyl to **3-OMT** in the non-phenolic ring worsens the affinity in derivative **1** even further and restores in derivative **2** the high affinity observed for tolcapone, accompanied by higher permeability and selectivity than **3-OMT,** hence of tolcapone (Table 4). The enhanced ability to penetrate the blood-brain barrier represents a major key point in developing a treatment for CNS TTR amyloidosis, such as familial leptomeningeal amyloidosis [59,60], as demonstrated by the studies attempting to increase the penetration both in the brain and in the periphery of tafamidis [61] and tolcapone itself [18].

## 3. Conclusions

Aided by preliminary docking analysis, using the scaffold of **3-OMT** as model, we chemically synthetised two compounds, **1** and **2**, having two methyl groups on the phenyl ring. The increased lipophilicity of **3-OMT** and its analogues resulted in an improved in vitro permeability through the blood-brain barrier, in vitro intestinal permeability, in vitro neuronal and hepatic safety, binding selectivity, and structural stabilisation with respect to tolcapone. **3-OMT**, **1**, and **2** all have a methoxy group in position 3 of the nitro phenyl ring that protects the molecule against the glucuronidation reaction occurring in the hepatic metabolic pathway, resulting in a lower toxicity than tolcapone. **3-OMT**, **1**, and **2** all show also a better oral bioavailability and plasma exposure, considering the improved in vitro intestinal permeability, with respect to the parent compound tolcapone. They have better TTR binding selectivity and a three-fold increase in blood-brain permeability compared to tolcapone.

Comparing the in vitro and in plasma properties of compound **2** to those of **3-OMT**, **1**, and the parent compound tolcapone, we can conclude that compound **2** has an overall better profile with higher TTR stabilization properties towards TTR, blood-brain permeability, optimal intestinal permeability with no efflux active transport, and an improved neuronal and hepatic safety. For these reasons, compound **2** could be selected as a potential lead drug candidate and progressed in a drug development program in the treatment for TTR amyloidosis, and prospectively for targeting CNS TTR amyloidosis like familial leptomeningeal and oculo-leptomeningeal amyloidosis.

## 4. Experimental Section

### 4.1. In Silico Structural and Docking Studies

To screen the affinity of the lipophilic analogues of **3-OMT**, we performed an automated molecular-binding recognition using the web-based docking server SwissDock “http://www.swissdock.ch/ (last access on 2 October 2020)”. The server is based on the docking algorithm EADock DSS [62,63] and clusters the most favourable binding modes according to their binding energy, using the algorithm FACT [64] We ran the pipeline three times in “accurate” and “blind” mode on each of the lipophilic candidates. As a control, we repeated the same docking on **3-OMT**, and verified the similarity of the binding mode with the deposited crystallographic structure (PDB ID: 6SUH) [39]. Docking runs were performed starting from the crystallographic structure of the TTR/3-OMT (PDB ID: 6SUH) deprived of the ligands [39]. Before the docking, the PBD files were modified to remove the solvent and the ligand, and to add the hydrogens. The position of the amino acids in the structure was randomized before running the simulation. To evaluate the hydrophobicity of each ligand, LogP values for tolcapone and its derivatives were computed by using Spartan Suite (Wavefunction, Inc. Spartan’16 (Irvine, CA, USA), Wavefunction, Inc.) and reported in Appendix A.

### 4.2. Chemistry: General

All moisture sensitive reactions were carried out under nitrogen or argon atmosphere, using previously degassed solvents. Solvents and reagents obtained from commercial sources were used without further purification. Analytical TLC were performed using prepared plates of silica gel (Merck 60 F 254 on aluminum, Rahway, NJ, USA) and then revealed with UV lights. Merck silica gel 60 was used for flash chromatography (40–63 µm). ^1^H-NMR and ^13^C-NMR spectra were recorded on a Bruker AV400 spectrometer (400 and 100 MHz as resonance frequency for ^1^H and ^13^C, respectively) and partially deuterated solvents were used as internal standards (δ values in ppm). All ^13^C-NMR spectra were performed with proton decoupling. Electrospray ionization (ESI) mass analyses were performed with a Waters single-quadrupole spectrometer and a LTQ Orbitrap XL spectrometer in positive mode using MeOH as solvent. Melting points were determined on a Gallenkamp apparatus in closed capillaries.

### 4.3. (4-(Benzyloxy)-3-methoxyphenyl)-1-(3,5-dimethylphenyl)methanol (***7***)

In a flame-dried three-necked round-bottomed flask, magnesium turnings (0.30 g, 12.52 mmol) were suspended in dry THF (3 mL) under argon atmosphere. Using a dropping funnel, a solution of 1-bromo-3,5-dimethylbenzene (2.31 g, 12.52 mmol) in dry THF (2.3 mL) was slowly added in 1 h. Then, a solution of 4-benzyloxy-3-methoxybenzaldehyde [47] **6** (2 g, 8.26 mmol) in dry THF (1.7 mL) was slowly added dropwise after 30 min into the reaction mixture under stirring. The reaction progress was monitored by TLC (eluent: hexane/ethyl acetate 7/3). After 1 h, the reaction was quenched by addition of saturated NH_4_Cl aqueous solution (5 mL) and extracted with Et_2_O (3 × 10 mL). The organic phase was washed with NaCl saturated solution (3 × 10 mL), dried with anhydrous Na_2_SO_4_, and evaporated at reduced pressure. The residue was crystallised from Et_2_O/petroleum ether to afford the product as white solid (1.97 g, 69%) of m.p. 94–95 °C. ^1^H NMR (DMSO-d_6_, 400 MHz): δ 7.48–7.26 (m, 5H, ArH); 7.01–6.89 (m, 4H, ArH); 6.87–6.72 (m, 2H, ArH); 5.68 (d, *J* = 3.9 Hz, 1H, OH); 5.53 (d, *J* = 3.9 Hz, 1H, CH); 5.03 (s, 1H, CH_2_); 3.74 (s, 3H, OCH_3_); 2.23 (s, 6H, CH_3_). ^13^C NMR (DMSO-d_6_, 100 MHz): δ 149.3 (Ar), 147.0 (Ar), 146.2 (Ar), 139.4 (Ar), 137.8 (Ar), 137.3 (Ar), 128.8 (Ar), 128.4 (Ar), 128.2 (Ar), 128.2 (Ar), 124.4 (Ar), 118.8 (Ar), 113.8 (Ar), 110.9 (Ar), 74.49 (CH), 70.4 (CH_2_), 56.0 (OCH_3_), 21.5 (CH_3_). MS (ESI, *m*/*z*): 371.14 ([M + Na]^+^).

### 4.4. (4-(Benzyloxy)-3-methoxyphenyl)-1-(3,5-dimethylphenyl)methanone (***8***)

t-BuONa (0.451 g, 3.61 mmol) and cyclohexanone (1.47 mL, 14.35 mmol) were added to a solution of **7** (1 g, 2.87 mmol) in toluene (4 mL). The mixture was stirred at reflux for 16 h. The reaction progress was monitored by TLC (eluent: hexane/ethyl acetate 7/3). The solution was cooled at 50 °C and then water (4 mL) was added. The organic phase was separated and the aqueous one was extracted with ethyl acetate (3 × 5 mL). The combined organic phases were washed with water (20 mL) and brine (20 mL) and, then, evaporated at reduced pressure obtaining an oily residue, which was crystallised with ethanol 96% to afford **3** as white powder (0.74 g, 75%) of m.p. 110.2–112.3 °C. ^1^H NMR (DMSO-d_6_, 400 MHz): δ 7.53–7.31 (m, 6H, ArH); 7.32–7.24 (m, 4H, ArH); 7.20 (d, *J* = 8.3 Hz, 1H, BnOCCH); 5.20 (s, 2H, CH_2_); 3.83 (s, 3H, OCH_3_); 2.35 (s, 6H, CH_3_). ^13^C NMR DMSO-d_6_, 100 MHz): δ 195.2 (C=O), 152.3 (Ar), 149.3 (Ar), 138.4 (Ar), 138.1 (Ar), 136.9 (Ar), 133.9 (Ar), 130.2 (Ar), 129.0 (Ar), 128.6 (Ar), 128.5 (Ar), 127.4 (Ar), 125.2 (Ar), 112.5 (Ar), 112.5 (BnOC*C*H), 70.4 (CH_2_), 56.0 (OCH_3_), 21.3 (CH_3_). MS (ESI, *m*/*z*): 347.12 ([M + H]^+^), 369.10 ([M + Na]^+^).

### 4.5. (4-Hydroxy-3-methoxyphenyl)-1-(3,5-dimethylphenyl)methanone (***9***)

To a solution in methanol (4.3 mL) of **8** (0.5 g, 1.44 mmol) and ammonium formate (0.36 g, 5.77 mmol), Pd/C 10% (catalytic amount) suspended in methanol (1.7 mL) was added. The reaction progress was monitored by TLC (eluent: hexane/ethyl acetate 7/3). The resulting mixture was stirred at reflux for 1 h. Subsequently, it was cooled in an ice/water bath and water (2 mL) and HCl 2M (0.4 mL) were slowly added up to slightly acidic pH. After addition of DCM (6 mL), the mixture was filtered through a celite pad. The organic phase was separated and the aqueous one was extracted with DCM (3 × 6 mL). The combined organic phases were washed with water (20 mL) and brine (20 mL) and dried with anhydrous Na_2_SO_4_. Solvent was evaporated under reduced pressure and product was obtained by crystallisation from DCM/petroleum ether as yellow crystals (0.30 g, 81%) of m.p. 130–131 °C. ^1^H NMR (CDCl_3_, 400 MHz): δ 7.53 (d, *J* = 1.9 Hz, 1H, H_a_); 7.39–7.33 (m, 3H, H_d_, H_c_); 7.22 (br s, 1H, H_e_); 6.97 (d, *J* = 8.2 Hz, 1H, H_b_); 6.08 (s, 1H, OH); 3.99 (s, 3H, OCH_3_); 2.40 (s, 6H, CH_3_). ^13^C NMR (CDCl_3_, 100 MHz): δ 196.0 (C=O), 150.0 (Ar), 146.6 (Ar), 138.4 (Ar), 137.8 (Ar), 133.5 (C-He), 130.2 (Ar), 127.5 (Ar), 126.2 (C-H_c_), 113.4 (C-H_b_), 111.7 (C-H_a_), 56.2 (OCH_3_), 21.3 (CH_3_) MS (ESI, *m*/*z*): 257.10 ([M + H]^+^), 279.07 ([M + Na]^+^).

### 4.6. (4-Hydroxy-3-methoxy-5-nitrophenyl)-1-(3,5-dimethylphenyl)methanone (***1***)

A mixture of glacial acetic acid (0.195 mL) and 65% nitric acid (0.140 mL) was slowly added to a suspension of **9** (0.25 g, 0.97 mmol) in glacial acetic acid (1.95 mL). The reaction was stirred at room temperature. The reaction progress was monitored by TLC (eluent: hexane/ethyl acetate 3/2). After 30 min, an ice/water mixture was added. The precipitate was filtered and crystallised by EtOH 96% to obtain yellow needle-shape crystal (0.087 g, 30%) of m.p. 144–145 °C. ^1^H NMR (DMSO-d_6_, 400 MHz): δ 7.75 (d, *J* = 2.0 Hz, 1H, H_a_); 7.57 (d, *J* = 2.0 Hz, 1H, H_b_); 7.35 (brs, 2H, H_c_); 7.32 (brs, 1H, H_d_); 3.95 (s, 3H, OCH_3_); 2.36 (s, 6H, CH_3_). ^13^C NMR (DMSO-d_6_, 100 MHz): δ (ppm) = 193.8 (C=O), 150.1 (Ar), 147.3 (Ar), 138.4 (Ar), 137.4 (Ar), 136.7 (Ar), 134.4 (Ar), 127.5 (Ar), 127.1 (Ar), 120.1 (Ar), 115.5 (Ar), 57.2 (OCH_3_), 21.2 (CH_3_). HRMS (ESI) calcd for C_16_H_15_NO_5_Na [M + Na]^+^ 324.0842, found 324.0840. HPLC (λ_270_) purity 98.07%, t_R_ 18.084.

### 4.7. (4-Hydroxy-3-methoxy-5-nitrophenyl)-1-(2,4-dimethylphenyl)methanone (***2***)

To a solution of 5-nitrovanillic acid [65] (0.20 g, 0.93 mmol) in DCM (3 mL) prepared in a freshly flame-dried two-necked round bottom flask, oxalyl chloride (0.16 mL, 1.86 mmol) was added, followed by a few drops of DMF. The reaction was stirred at room temperature and its progress monitored by TLC (eluent: hexane/ethyl acetate 1/1). After 1 h, verified the reaction completion, the solvent, and all traces of oxalyl chloride were removed under vacuum. The residue was dissolved in a mixture of dry DCM (4 mL) and m-xylene (4 mL). The solution was cooled in an ice/water bath and AlCl_3_ (0.16 g, 1.20 mmol) was added. The resulting suspension was kept stirred for 16 h at rt. The reaction progress was monitored by TLC (eluent: hexane/ethyl acetate 1/1). The reaction was quenched by evaporation at reduced pressure of the volatiles. The residue was dissolved in DCM (5 mL), washed with HCl 1M (3 × 5 mL) and water (2 × 5 mL) and basified with 30% ammonia solution (pH 9). The precipitate was filtered and suspended in HCl 1 M (10 mL). DCM (10 mL) was added to dissolve the solid and the organic phase was separated, dried over anhydrous Na_2_SO_4_ and evaporated under reduced pressure to obtain the product as yellow powder (0.201 g, 57%) of m.p. 126–127 °C. ^1^H NMR (CD_3_OD) δ 7.81 (d, *J* = 1.9 Hz, 1H, H_a_), 7.72 (d, *J* = 1.9 Hz, 1H, H_b_), 7.25 (d, *J* = 7.8 Hz, 1H, H_c_), 7.21 (br s, 1H, H_e_), 7.15 (d, *J* = 7.3, 1H, H_d_), 4.00 (s, 3H, OCH_3_), 2.42 (s, 3H, CH_3_), 2.31 (s, 3H, CH_3_). ^13^C NMR (CD_3_OD) δ 196.0 (C=O), 150.2 (Ar), 141.1 (Ar), 136.8 (Ar), 135.2 (Ar), 134.7 (Ar), 131.6 (Ar), 128.4 (Ar), 128.1 (Ar), 125.7 (Ar), 120.1 (Ar), 114.5 (Ar), 57.0 (OCH_3_), 21.8 (CH_3_), 19.1 (CH_3_). HRMS (ESI) calcd for C_16_H_15_NO_5_Na [M + Na]^+^ 324.0842, found 324.0845. HPLC (λ_270_) purity 95.58%, t_R_ 17.685.

### 4.8. Key Compounds Purity

Tolcapone and 3-O-methyltolcapone, both purchased, had a minimum purity of 97.0 and 98.0%, respectively, as indicated by the vendor; **1** and **2** had a purity of 98.0 and 95.6%, respectively, as determined by HPLC (see Appendix A).

### 4.9. Chromatographic Hydrophobicity Index (CHI)

The assay is a chromatographic fast gradient reversed-phase method which allows to characterize the lipophilicity of compounds in high throughput screening regime at three different pH values (2.0; 7.4; 10.5). The so defined CHI is obtained from the gradient (buffer/Acetonitrile) retention time after calibrating the chromatographic system with a test mixture of ten known standard compounds. For most compounds, CHI is between 0 and 100 and it approximates the percentage (by volume) of acetonitrile required to achieve an equal distribution of compound between the mobile and the lipophilic stationary phase of a C18 chromatographic column. Compounds that are not retained on the column are assigned to a CHI value < 0, which means very low lipophilicity. For highly lipophilic compounds, which are strongly retained by the column, a CHI value of >100 is assigned. For any value in between, the actual CHI value is calculated based on the calibration line of the reference compounds [49].

### 4.10. Protein Expression and Purification

The pET11-b vector containing the human TTR wild type was transformed into *E. coli* BL21 cells and grown in simplified LB autoinduction medium supplemented with 100 µg/mL ampicillin at 37 °C. At an absorbance value of 600 nm around 0.3, the temperature was shifted to 30 °C. Cells were grown overnight (16–18 h) and harvested by refrigerated centrifugation for 20 min at 5000 g.

Cell pellets were resuspended in lysis buffer (50 mM sodium phosphate, 300 mM NaCl, 0.5 mM ethylenediaminetetraacetic acid (EDTA), 0.1% β-mercaptoethanol pH 8) and lysed by sonication with a total sonication energy input of about 8315 J per 15 mL cell suspension volume and a short sonication-cooling cycle (10 s–10 s). Cell debris was removed by refrigerated centrifugation at 12,000× *g* for 20 min at 4 °C. Ammonium sulphate powder was added to the supernatant to reach 55% saturation at 0 °C and incubated for 2 h on ice before subjected to another cycle of centrifugation to separate the precipitation. The supernatant was filtered and applied to a RESOURCE™ PHE (phenyl) 1 mL column (Cytiva, Marlborough, MA, USA) previously equilibrated with 50 mM Tris, 1.5 M ammonium sulphate pH 7.5. Proteins were eluted with a linear gradient of ammonium sulphate to 0 M. The selected fractions were diluted or dialyzed against 50 mM Tris pH 7.5 and loaded onto an anion exchange column, RESOURCE™ Q 1 mL (Cytiva). Proteins were eluted with a linear NaCl gradient from 0 M to 0.5 M. The protein fractions were concentrated and finally subjected to Superdex 75 10/300 size exclusion chromatography column (GE Healthcare) equilibrated with 50 mM Tris, 100 mM NaCl pH 7.5. TTR tetramer is approximately 55 kDa with extinction coefficient of 77,600 M^−1^ cm^−1^. Protein was calculated as a tetramer in all experiments unless specified otherwise.

### 4.11. Solubility (Kinetic) and Blood Brain Barrier Permeability (PAMPA-BBB)

The assay combines solubility and a non-cell-based Blood Brain Barrier permeability assay (Parallel Artificial Membrane Permeation Assay; PAMPA-BBB) into a single workflow in which a portion of the filtrate from the Solubility filter plate is added to the donor compartment of the MultiScreen filter plate for the PAMPA-BBB analysis. The aqueous solubility was estimated in duplicate at pH 7.4 in Phosphate Buffered Saline (PBS) by mixing, incubating, and filtering a solution (1% DMSO) in a solubility filter plate (polycarbonate filter). Solutions were filtered into a 96-well collection plate using vacuum filtration and then analysed by UV/Vis spectroscopy. The relative solubility was then calculated using the sum of the recorded values as compared to a standard calibration curve. The solubility classification ranges low (S < 10 µM), moderate (10 < S < 100 µM) and high (S > 100 µM). PAMPA-BBB determination was carried out in a 96-well MultiScreen Permeability plate and the ability of a compound to diffuse from a Donor to an Acceptor compartment separated by a Porcine Brain Lipid Extract (PBLE) dissolved in dodecane (20 mg/mL) layer on a PVDF membrane support was measured. The assay was performed in duplicate in PBS at pH 7.4 and the analysis is carried out by UV/Vis spectroscopy.

The compounds are classified into 2 types: CNS+ compounds that can cross the BBB and CNS- compounds that cannot according to the Pe values (BBB−, Pe < 2 × 10^−6^ cm/s; BBB +/−, 2 × 10^−6^ cm/s < Pe < 4 × 10^−6^ cm/s; BBB+, Pe > 4 × 10^−6^ cm/s).

Membrane retention is also evaluated according to Equation (1),
(1)$$1-R_{m}=\frac{C_{Acc}V_{Acc}+C_{Donor}V_{Donor}}{Sol\times V_{Donor}}$$
where *Rm* is the membrane retention factor. 1 − *Rm* between 1.2 and 0.8 is considered fully acceptable for the reliability of the results.

### 4.12. Caco-2 Intestinal Permeability

Caco-2 cells (cod. HTB-37) were bought from ATCC (American Type Culture Collection, Rockville, MD, USA). Caco-2 cells were cultured for 21 days on a 24 trans-well plate to reach the right confluence and differentiate until the formation of the intestinal epithelium mimetic monolayer. The test substance permeability was measured in both directions (apical to basolateral [A→B] and basolateral to apical [B→A]) across the cell monolayer. The assay was carried out in HBSS transport medium at pH 7.4. Test substance at single concentration (10 μM) was tested in duplicate at one time point (60 min). Monolayer integrity of Caco-2 cells was evaluated by TEER measurements during the experiment and by Lucifer Yellow test after the incubation with the test compound. The concentrations of the test substance and/or reference controls were measured by LC-MS/MS. Apparent Permeability (Papp) is reported in nm/s and test substance permeability is classified as low permeability (Papp < 100 nm/s), moderate permeability (100 nm/s < Papp < 300 nm/s), high permeability (Papp > 300 nm/s).

The efflux ratio (Papp BA/Papp AB) was determined, providing an Indicator of the compound’s active efflux. A P-glycoprotein (P-gp) inhibitor, Elacridar (EL 2 µM), was included to identify the active transport mediated by apical efflux transporters. When the inhibitor leads the efflux ratio to a significant reduction (>50%) or to unity, the test compound is classified as a probable substrate of that efflux transporter. As control, the modulation of the efflux ratio of a reference known substrate (Digoxin) was evaluated with or without the respectively inhibitors in every experiment. Atenolol and Metoprolol were used as low- and high-passive permeability controls, respectively. Table 4 reports data for the control compounds. In parallel, mass balance (%) of recovered test substance was also determined. A recovery% within 70–120 is considered fully acceptable whereas permeability data with mass balance in ranges 50–70% or 121–150% should be taken with caution. Outside these ranges the permeability data is not reliable.

### 4.13. Cytotoxicity in Neuronal-Derived SH-SY5Y and Hepatic-Derived Hepg2 Human Cell Lines: (ATP Viability Assay)

The neuroblastoma cell line SHSY-5Y (ATCC CRL2266) was cultured in EMEM ATCC medium supplemented with 10% FBS (Euroclone CAT No. ECS0180D), 100 units/mL penicillin, and 100 ug/mL streptomycin sulphate (Euroclone CAT No. ECB3001). Cells were grown at 37 °C in an atmosphere of 5% CO_2_ and plated at 10,000 cells/well in a 96-well plate (Corning Cat. N. 3610). The compounds were dissolved at a concentration of 10 mM in DMSO. Cells were stimulated for 48 h at 37 °C with serial dilutions of the compounds at the following concentrations: 0.1, 1, 10, 50, 100, and 200 μM and dispensed with an HP Dispenser (Tecan), as triplicates for each concentration. DMSO normalisation at 1.5% was performed in each well.

The human liver cancer cell line HepG2 (ATCC HB-8065) was cultured in MEM ATCC medium supplemented with 10% FBS (Euroclone CAT No. ECS0180D), 100 units/mL penicillin, and 100 ug/mL streptomycin sulphate (Euroclone CAT No. ECB3001). Cells were grown at 37 °C in an atmosphere of 5% CO_2_ and plated at 10,000 cells/well in a 96-well plate (Corning Cat. N. 3610). Compounds were dissolved at the concentration of 50 mM in DMSO.

Cells were stimulated for 48 h at 37 °C with serial dilutions of the compounds at the following concentrations: 0.1, 1, 10, 50, 100, 200, 300, and 500 μM and dispensed with an HP Dispenser (Tecan), as triplicates for each concentration. DMSO normalisation at 1.0% was performed in each well. Lysis buffer was used as a positive control of dead cells with tamoxifen (Sigma-Aldrich CAT No. T5648).

Cell viability was evaluated by measuring ATP levels using CellTiter Glo (Promega Cat.n. G7570) according to manufacturer’s instructions. A four-parameter nonlinear regression was used (GraphPad Prism 5 (version 5.04) of GraphPad Software Inc.) to calculate the half maximal effective concentration (EC 50) considering 100% living cells (cells with DMSO) and 0% living cells (cells treated with Lysis buffer 0.01%).

### 4.14. Isothermal Titration Calorimetry (ITC)

ITC experiments were carried out using an ITC200 microcalorimeter (MicroCal/Malvern) with 3 or 5 µM TTR and 100 µM ligand both in 1× buffer A (25 mM HEPES pH 7,4, 100 mM KCl, 1 mM EDTA) and 5% DMSO at 25 °C. 1.5-μL aliquots of ligand solution were injected into the protein solution at 25 °C (time interval 180 s, stirring at 750 rpm). Data were fitted using the “one-binding-site model” of the Malvern version of ORIGIN v.7. The heat of binding (ΔH), the stoichiometry (n), entropy (ΔS) and the dissociation constant (Kd) were then calculated from plots of the heat evolved per mole of ligand injected versus the molar ratio of ligand to protein using the software provided by the vendor (Table 3). Sample preparation is detailed below. A concentrated TTR solution (∼140 µM) was dialyzed thoroughly in 2× buffer A overnight at 4 °C; protein was then collected, centrifuged (10′ at 12,000 rpm, 4 °C) and its concentration evaluated by UV/vis absorbance spectrum. A 10 µM protein stock in 2× buffer was then prepared and diluted 1:1 with a 10% DMSO solution, to obtain a 5 µM TTR solution in 1× buffer A and 5% DMSO. Different protein concentrations (3–5 µM) assayed in the replicates were prepared accordingly, starting from a 2× protein solution in 2× buffer A. Each ligand was dissolved in 100% DMSO and a 2 mM stock solution in 100% DMSO was prepared. A 200 µM compound solution was obtained in 10% DMSO and diluted 1:1 with 2× buffer A, to obtain a 100 µM ligand solution to be used for the ITC assays (in 1× buffer A and 5% DMSO). 

### 4.15. Stability Studies of TTR in Serum by Immunoblotting

The TTR tetramer stability assay in the presence of urea and TTR ligands was performed according to a previously reported Western Blot procedure [39], with some modifications as follows. Aliquots of human plasma were diluted 8-fold in sodium phosphate buffer (NaP 40 mM, NaCl 150 mM pH 7.4) supplemented with 2 µM or 20 µM of each ligand. A plasma aliquot supplemented with DMSO was used as a negative control. After 2 h of incubation at 20 °C, an equal volume of 8 M urea in sodium phosphate buffer was added to obtain a final 4 M urea concentration and final ligand concentration of 1 µM or 10 µM. Incubation was prolonged for 18 h at 20 °C, followed by non-denaturing SDS-PAGE using Tris-Glycine buffers containing 0.025% of SDS in the running buffer and 0.2% of SDS in the Laemmli buffer. As previously reported [66], SDS at this concentration does not denature TTR tetramer but does prevent re-association of TTR monomer. Blotting was accomplished with a Trans-Blot SD transfer apparatus (BIORAD), followed by incubation of nitrocellulose membranes with blocking buffer containing 5% skim milk overnight at 25 °C. Immunodetection of TTR monomers was performed by employing rabbit anti-human TTR polyclonal antibody (Dako) as a primary and goat anti-rabbit antibody labelled with DyLight 680 (SERACARE) as a secondary. Western Blot images were recorded by using ChemiDoc MP imaging system (BIORAD) and analysed with Image Lab software v. 5.1 (BIORAD). The experiment was performed in triplicate, and ligands stabilisation capability was evaluated as the relative intensity of the TTR monomer in each condition in comparison to the negative control. Statistical analyses by means of ANOVA and graphic representations were performed with GraphPad Prism.

### 4.16. Crystallisation

Purified human TTR at the concentration of ~5 mg/mL was screened by micro batch under oil technique using commercial crystallisation screens from Molecular Dimension in 96 well plates. The condition that gave the crystals, 0.2 M calcium acetate hydrate, 0.1 M Tris 7.5, and 25% *w*/*v* PEG 2000 MME, was further optimized by the vapor diffusion method and used for the diffraction experiment. Ligands were incubated with protein overnight at 4 °C at a molar ratio of protein monomer to ligand of 1:5 or 1:10 prior crystallisation. The crystals were flash frozen in mother liquor containing 12% glycerol or 40% CryoMixTM 2 (25% *v*/*v* Diethylene glycol, 25% *v*/*v* Glycerol, and 25% *v*/*v* 1,2-Propanediol) (Molecular Dimension).

### 4.17. Data Collection and Crystal Structure Determination

X-ray data were collected at a cryogenic temperature at beamline P13 operated by EMBL Hamburg at the PETRA III storage ring (DESY, Hamburg, Germany) [67]. Data reduction was carried out by XDS [68]. Scaling was performed by Aimless [69]. The protein structure was determined by molecular replacement using MrBump [70]. The structure was refined using the PHENIX suite [71] and built in COOT [72]. Data collection and refinement statistics are shown in Appendix A. The crystallographic refinement data and final PDB coordinates for the TTR/**1** and TTR/**2** were deposited in the Protein Data Bank with accession codes 8C85 and 8C86, respectively.

## Figures and Tables

**Figure 1 ijms-25-00479-f001:**
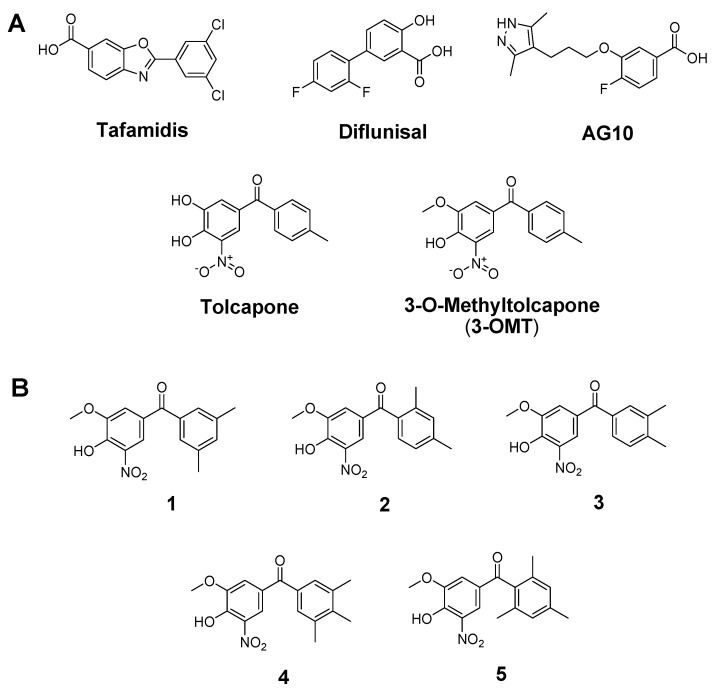
(**A**) Chemical structures of TTR amyloidosis inhibitors and of 3-O-methyltolcapone (**3-OMT**); (**B**) Lipophilic analogues of **3-OMT** assessed here with docking studies. Compound labels are indicated as numbers in bold.

**Figure 2 ijms-25-00479-f002:**
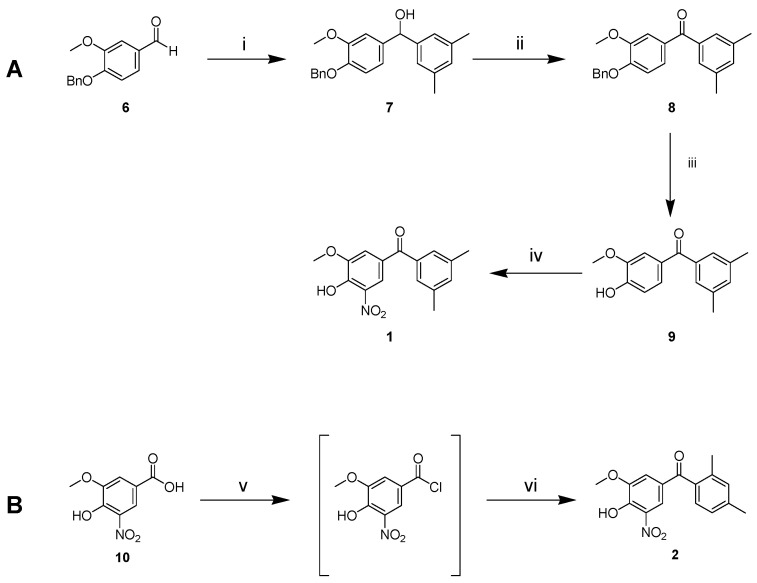
(**A**) Synthesis of **1**. Reaction conditions: (i) magnesium turnings, 3,5-dimethylbromobenzene, dry THF, rt, 1.5 h; (ii) tBuONa, cyclohexanone, toluene, reflux, 16 h; (iii) HCOO^−^NH_4_^+^, Pd/C, MeOH, reflux, 0.5 h; (iv) glacial AcOH, 65% HNO_3_, rt, 0.5 h. (**B**) Synthesis of **2**. Reaction conditions: (v) oxalyl chloride, dry DMF, dry DCM, rt, 1 h; (vi) AlCl_3_, dry DCM, dry m-xylene, rt, 16 h.

**Figure 3 ijms-25-00479-f003:**
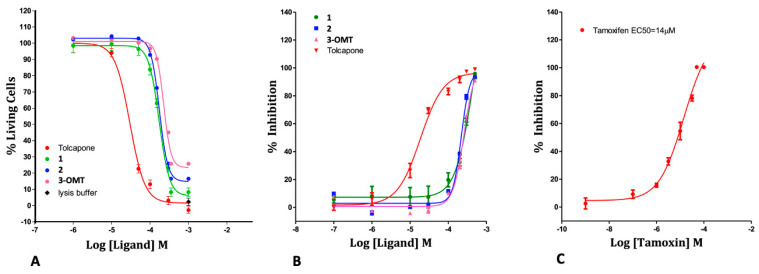
(**A**) Cytotoxicity in Neuroblastoma SH-SY5Y cell line (ATP Viability Assay). (**B**) Cytotoxicity in Hepatocarcinoma HepG2 cell line (ATP Viability Assay). (**C**) Positive control (Tamoxifen) of the HepG2 assay. The full dose response curves at the tested concentrations are shown as the average and standard error for each concentration. Percentage of living cells were calculated, and four-parameter nonlinear regression curve was generated using GraphPad Prism 5 (version 5.04) of GraphPad Software Inc. (San Diego, CA, USA), to calculate EC_50_ values.

**Figure 4 ijms-25-00479-f004:**
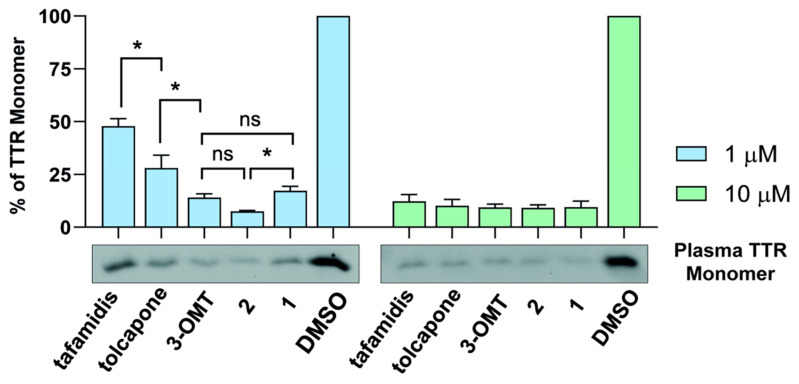
Binding selectivity and structural stabilisation of human TTR in diluted plasma samples, by tolcapone, **3-OMT**, **1**, and **2**. Histograms represent the relative abundance of TTR monomer after supplementation of tolcapone analogues at two different concentrations (1 µM, left and 10 µM, right), in comparison to the negative control (DMSO). Data are expressed as mean (SD) of three independent experiments, and a representative western blot result is shown below each graph. Data were analysed by one-way ANOVA and symbols indicate statistical difference between treatments: *p* > 0.05 (ns), *p* ≤ 0.05 (*). Only the most relevant comparisons are displayed. Values (%) at concentrations 1 µM are: tafamidis, 47.8 ± 3.6; tolcapone, 28 ± 6.2; **3-OMT**, 14.1 ± 1.8; **2**, 7.5 ± 0.4; **1**, 17.2 ± 2.1; DMSO, 100 ± 0. Values (%) at concentrations 10 µM are: tafamidis, 12.3 ± 3.2; tolcapone, 10.2 ± 3; **3-OMT**, 9.4 ± 1.5; **2**, 9.2 ± 1.5; **1**, 9.5 ± 2.8; DMSO, 100 ± 0.

**Figure 5 ijms-25-00479-f005:**
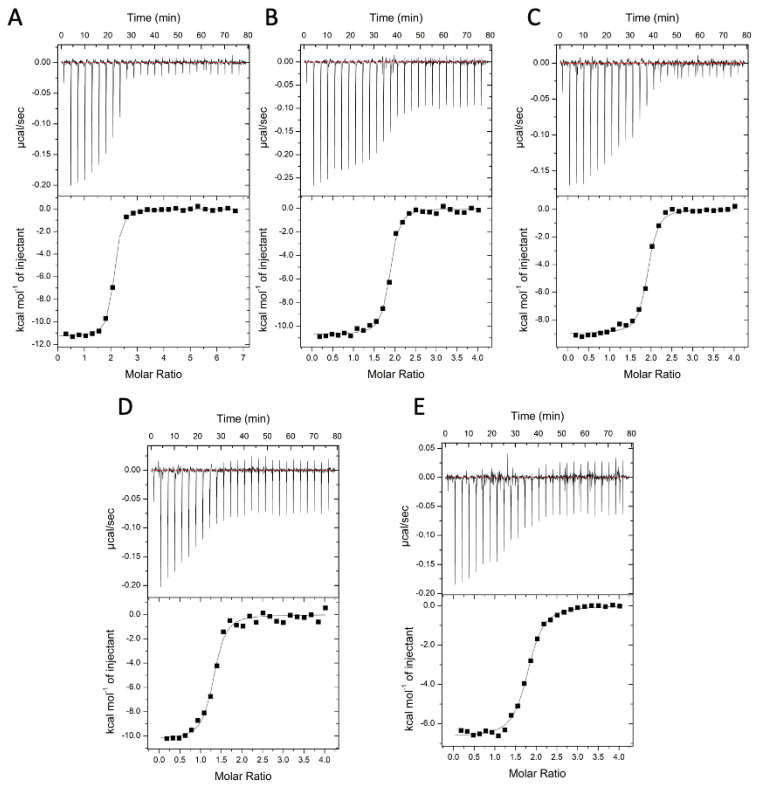
Raw ITC binding data of the selected inhibitors to TTR. Analysis by ITC measurements was performed by titrating 3 or 5 µM TTR protein solution with (**A**) tolcapone, (**B**) **2**, (**C**) **3-OMT**, (**D**) **1**, or (**E**) tafamidis, in 1× buffer 25 mM Hepes pH 7.4, 100 mM KCl, 1 mM EDTA and 5% DMSO.

**Figure 6 ijms-25-00479-f006:**
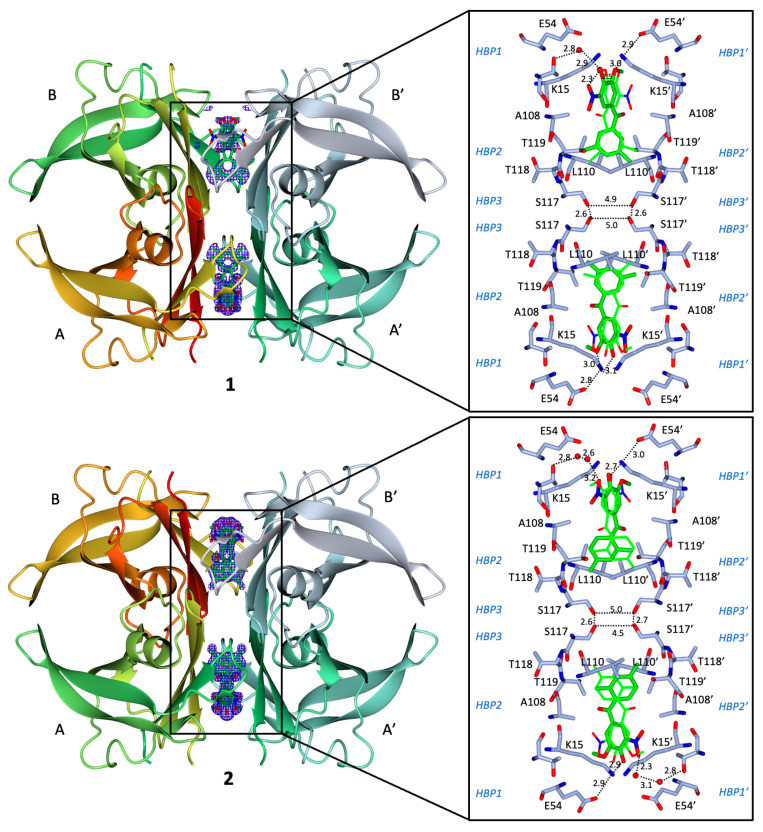
TTR−ligand interactions in the crystal structures of the **1** (PDB ID: 8C85) and **2** (PDB ID: 8C86) TTR complexes. (**Left**) OMIT (F_o_–F_c_) Fourier difference electron density maps (blue), contoured at 3σ level, of tolcapone analogues (top: **1**; bottom: **2**) bound in the two cavities of the TTR tetramer formed by monomers A and B, and symmetry related A’ and B’. (**Right**) Detailed views of the two symmetry-related binding modes of each ligand (dark green) within the T4 binding cavity, interacting through residues present in HBP1/HBP1′ (Lys15, Glu54, and Thr106), HBP2/HBP2′ (Leu110, Leu17, Lys15), and HBP3/HBP3′ (Ser117, Leu110, Ala108, and Thr119). Interacting residues and ligands are represented as sticks. Symmetry related image of the ligands is represented as fine bonds. The carbon, nitrogen and oxygen are light blue (green for the ligands), blue and red, respectively. Waters are represented as red spheres. H-bonds are shown as dotted lines with distances indicated in Å.

**Figure 7 ijms-25-00479-f007:**
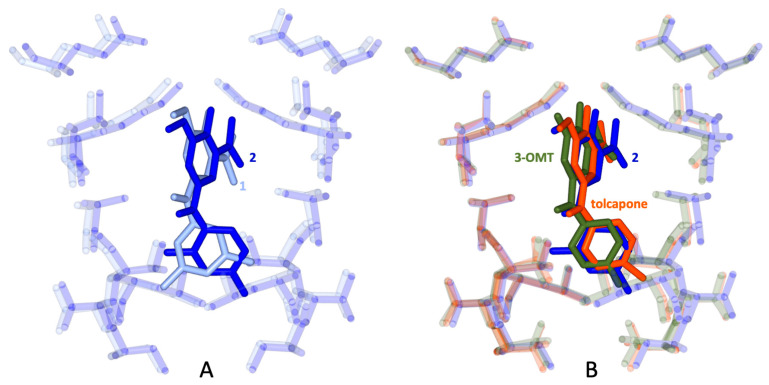
Superposition of the pose within the binding site of TTR of compound **2** (colored in blue) with: (**A**) compound **1** (light blue); (**B**) **3-OMT** (green) and tolcapone (red).

**Table 1 ijms-25-00479-t001:** CHI values of the four tested compounds.

	CHI (pH 2.0)	CHI (pH 7.4)	CHI (pH 10.5)
tolcapone	78.2	n.d.	49.3
**3-OMT**	89.2	50.2	46.0
**1**	97.8	55.5	51.7
**2**	96.1	53.8	50.2

**Table 2 ijms-25-00479-t002:** Toxicity in neuronal-derived SH-SY5Y and hepatic-derived HepG2 human cell lines of tolcapone, **3-OMT**, and its lipophilic analogues **1** and **2**.

Compound	SH-SY5Y EC_50_ (µM)	HepG2 EC_50_ (µM)
tolcapone	29.8 ± 1.1	17.5 ± 2.4
**3-OMT**	226.3 ± 1.3	262.3 ± 20.7
**1**	172.1 ± 5	410 ± 33
**2**	174.4 ± 3.7	215.6 ± 2.2

**Table 3 ijms-25-00479-t003:** Parameters for binding isotherms, fitted using a single-binding-site model (Microcal Origin v.7 software). The integrated energy values were normalised for injected protein.

Ligand	n	K_d_ (nM)	∆H (kcal mol^−1^)	∆S (kcal mol^−1^ T^−1^)
tafamidis	2.4 ± 0.1	128 ± 63	−3.2 ± 0.3	20.9 ± 2.1
tolcapone	2.0 ± 0.1	26 ± 4	−11.8 ± 0.5	−4.8 ± 1.9
**3-OMT**	1.9 ± 0.2	33 ± 9	−8.9 ± 0.3	4.6 ± 0.5
**1**	1.8 ± 0.1	71 ± 26	−11.5 ± 1.3	−2.9 ± 0.9
**2**	2.1 ± 0.3	25 ± 5	−10.5 ± 0.2	−0.4 ± 0.1

**Table 4 ijms-25-00479-t004:** Caco-2 Intestinal Permeability. Expected values are reported in brackets.

Compound	Direction/±Inhibitor	P_app_ (nm/s) [Expected Values]	Recovery %	Efflux Ratio (Reduction %)
Atenolol †	A→B	<10 [<10]	86 ± 1	
Metoprolol †	A→B	>400 [>400]	105 ± 5	
Digoxin † (P-gp substrate)	A→B	2 ± 1 [2]	67 ± 5	
	B→A	141 ± 26 [139]	71 ± 6	>10
	A→B + EL * 2 µM	35 ± 5 [26]	72 ± 1	
	B→A + EL * 2 µM	83 ± 31 [63]	75 ± 3	2.3 (96%)
tolcapone	A→B	101 ± 12	53 ± 1	2
	B→A	206 ± 58	70 ± 4	
**3-OMT**	A→B	160 ± 29	66 ± 4	1.5
	B→A	243 ± 30	83 ± 8	
**1**	A→B	240 ± 21	64 ± 4	1
	B→A	231 ± 23	77 ± 2	
**2**	A→B	250 ± 3	69 ± 8	1
	B→A	268 ± 25	80 ± 9	

† Reference compounds, * EL: Elacridar.

**Table 5 ijms-25-00479-t005:** Determined solubility (Kinetic) and Blood-Brain Barrier Permeability (PAMPA-BBB) values for control molecules (verapamil, caffeine, and theophylline) and tested compounds (tolcapone, **3-OMT**, **1**, and **2**). Expected values are reported in brackets.

Compound	BBB-Pe (10^−6^ cm/s) [Expected Values]	1-Rm	CNS Class [Expected Values]	Solubility (µM)
Verapamil †	12.0 (±2.4) [>10]	0.8	CNS+ [CNS+]	>500
Caffeine †	1.9 (±0.1) [1.3]	0.9	CNS− [CNS−]	>500
Theophylline †	0.2 (±0.1) [0.12]	1.0	CNS− [CNS−]	>500
tolcapone	5.6 (±1.2)	0.9	CNS+	>500
**3-OMT**	14.4 (±0.2)	0.9	CNS+	>500
**1**	15.3 (±2.0)	0.9	CNS+	>500
**2**	13.5 (±3.9)	1.0	CNS+	>500

† Reference compounds.

## Data Availability

Data contained within the article or Appendix A.

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
