# Peer review of "3-O-Methyltolcapone and Its Lipophilic Analogues Are Potent Inhibitors of Transthyretin Amyloidogenesis with High Permeability and Low Toxicity"

_ijms, 2023, doi:10.3390/ijms25010479_

Round 1

Reviewer 1 Report

Comments and Suggestions for Authors

In this study, Poonsiri and colleagues report two novel 3-O-methyltolcapone lipophilic analogues as inhibitors of TTR amyloidogenesis. The following should be addressed prior to the publication of the paper.

-Line 140 and 168: Please remove extra “.” at the end of each heading.

-Line 162: Please further clarify the rationale for the choice of adding a variable number of methyl groups for increasing compound lipophilicity.

-Line 183-195: Please consider presenting a figure showing each compound docked pose in the supplementary document for better clarity.

-Figure 6: Please increase the resolution of each figure to be more clearly visible.

-Figure 7: Please consider including the residue labels in the figures and adjust them to be clearly visible.

-Line 485: (a typical comment for the entire paper) Please consider using "PDB ID" consistently before any PDB code with appropriate referencing.

Author Response

In this study, Poonsiri and colleagues report two novel 3-O-methyltolcapone lipophilic analogues as inhibitors of TTR amyloidogenesis. The following should be addressed prior to the publication of the paper.

-Line 140 and 168: Please remove extra “.” at the end of each heading.

DONE IT.

-Line 162: Please further clarify the rationale for the choice of adding a variable number of methyl groups for increasing compound lipophilicity.

(now line 220) We added the sentence “Nonpolar groups (e.g., methyl) are tipically added to a molecule to enhance lipophilicity “ and its reference which deals with the rationale:

https://www.sciencedirect.com/science/article/abs/pii/B9780128010761000101

-Line 183-195: Please consider presenting a figure showing each compound docked pose in the supplementary document for better clarity.

Among the compounds presented in Figure 1, panel B, compound 4 and 5 did not pass the docking studies. The poses of 1, 2 and 3 are were experimentally determined and presented in figure 6 and 7. The docking poses of 1 and 2 are in agreement with the experimental results.

We added two sentences on page 5, line 259, “The docking poses of analogues 1 and 2 were later experimentally confirmed.“ and on page 15, line 477 “The experimentally determined poses of 1 and 2 are in agreement with the results of the docking studies.” (line 159)

-Figure 6: Please increase the resolution of each figure to be more clearly visible.

DONE IT.

-Figure 7: Please consider including the residue labels in the figures and adjust them to be clearly visible.

DONE IT.

-Line 485: (a typical comment for the entire paper) Please consider using "PDB ID" consistently before any PDB code with appropriate referencing.

DONE IT.

We thank the referee for the constructive comments and the support to our work.

Reviewer 2 Report

Comments and Suggestions for Authors

In this study, authors have synthesized analogs of an existing drug, tolcapone, in order to develop inhibitors of c, with more favorable properties.

Overall, this is a well-thought-out and well-conducted study. The introduction was well written, covered the main background points and led up to the aim of the study. The authors gave a comprehensive description of the used methods which would allow the easy replication of the study. All the data from experiments described in the methods section are reported and the results were adequately interpreted and given the broader context.

However, some minor issues should be addressed by the authors before the manuscript could be published:

- CHI value could not be determined for the tolcapone at the physiological pH, which limits certainty of the statement in lines 233-235 that derivatives have overall higher hydrophobicity. This should be discussed as a limitation. Ideally, authors should have experimentally determined the log P value (or used another experimental method to determine lipophilicity) of all compounds of interest.

- line 526: Authors in the first line of Conclusions wrote: “Aided by preliminary molecular dynamics simulation…” However, Molecular Dynamics was not used in this study, only docking analysis. Molecular dynamics and docking are related but distinct computational techniques. The authors themselves stated, in lines 173-175, that they only used docking, as an alternative to Molecular Dynamics.

- Sentence in lines 534-536: “The methoxy group in position 3 of the nitro phenyl ring…” is incomprehensible and should be rewritten.

Overall, this is a well-written manuscript with appropriate methodology, that offers a valuable contribution to the field of transthyretin-related amyloidosis treatment.

Comments on the Quality of English Language

Minor issues were detected.

Author Response

In this study, authors have synthesized analogs of an existing drug, tolcapone, in order to develop inhibitors of c, with more favorable properties.

Overall, this is a well-thought-out and well-conducted study. The introduction was well written, covered the main background points and led up to the aim of the study. The authors gave a comprehensive description of the used methods which would allow the easy replication of the study. All the data from experiments described in the methods section are reported and the results were adequately interpreted and given the broader context.

However, some minor issues should be addressed by the authors before the manuscript could be published:

- CHI value could not be determined for the tolcapone at the physiological pH, which limits certainty of the statement in lines 233-235 that derivatives have overall higher hydrophobicity. This should be discussed as a limitation. Ideally, authors should have experimentally determined the log P value (or used another experimental method to determine lipophilicity) of all compounds of interest.

AGREED. We changed the line 233-235 from OLD: "In summary, it was found that the overall hydrophobicity of the three derivatives 3-OMT, 1 and 2 is higher than that of tolcapone" to:

NEW: "However considering the condition at acidic pH, where the molecules should be ideally neutral, the CHI values indicates that the intrinsic hydrophobicity of the three derivatives 3-OMT, 1 and 2 is higher than that of tolcapone."

- line 526: Authors in the first line of Conclusions wrote: “Aided by preliminary molecular dynamics simulation…” However, Molecular Dynamics was not used in this study, only docking analysis. Molecular dynamics and docking are related but distinct computational techniques. The authors themselves stated, in lines 173-175, that they only used docking, as an alternative to Molecular Dynamics.

GOOD POINT. AGREED. Changed “molecular dynamics” to “docking analysis”.

- Sentence in lines 534-536: “The methoxy group in position 3 of the nitro phenyl ring…” is incomprehensible and should be rewritten.

GOOD POINT. AGREED. ∫Changed “The methoxy group in position 3 of the nitro phenyl ring…” to “3-OMT, 1 and 2 all show also a better oral bioavailability and plasma exposure”.

Overall, this is a well-written manuscript with appropriate methodology, that offers a valuable contribution to the field of transthyretin-related amyloidosis treatment.

We thank the referee for the constructive comments and the support to our work.

Reviewer 3 Report

Comments and Suggestions for Authors

In the submitted manuscript (ijms-2780146), the authors examined the biological activity and pharmacological properties of 3-O-methyltolcapone and its lipophilic (newly synthesized) analogs as possible inhibitors of transthyretin (TTR) amyloidogenesis of importance because Parkinson's disease is recently being considered as a type of amyloidosis. Based on the results of comprehensive docking studies and then (for the best candidates) in vitro and ex vivo tests, they convincingly showed that compounds of interest are indeed potent inhibitors of transthyretin amyloidogenesis (the high degree of TTR stabilization), with favorable permeability (they effectively cross the blood-brain barrier) and low toxicity (improved neuronal and hepatic safety). Those results suggest the relevance of examined compounds as potent inhibitors of TTR amyloidogenesis, referring to the continuation and expansion of the research.

Overall, it was a pleasure to read the text of the submitted paper, considering the clear concept, chosen methodology, adequately processed and presented a large number of results, excellent writing style, and language, and, equally importantly, very nicely presented and discussed results, with guidelines for further work in this exciting area of research. Kudos to the authors.

I have no specific complaints except for those that can be classified as technical imperfections.

Please put all Latin expressions in italics (e.g., via), standardize the writing of the expression in vitro (there is no need to have a hyphen in between, i.e., to write in-vitro), correct minor typographical errors (e.g., an extra comma in a sentence on line 132, the mark for micro on line 753, as well as the abbreviation for the catalog number in the same place in the text (e.g., put CAT No.), or the space between the number and the measure mark: one example is on line 778). I also believe that reference number 26 is not entirely correctly cited (is it this article?: https://www.cadth.ca/sites/default/files/cdr/clinical/sr0603-tegsedi-clinical-review-report .pdf).

Author Response

In the submitted manuscript (ijms-2780146), the authors examined the biological activity and pharmacological properties of 3-O-methyltolcapone and its lipophilic (newly synthesized) analogs as possible inhibitors of transthyretin (TTR) amyloidogenesis of importance because Parkinson's disease is recently being considered as a type of amyloidosis. Based on the results of comprehensive docking studies and then (for the best candidates) in vitro and ex vivo tests, they convincingly showed that compounds of interest are indeed potent inhibitors of transthyretin amyloidogenesis (the high degree of TTR stabilization), with favorable permeability (they effectively cross the blood-brain barrier) and low toxicity (improved neuronal and hepatic safety). Those results suggest the relevance of examined compounds as potent inhibitors of TTR amyloidogenesis, referring to the continuation and expansion of the research.

Overall, it was a pleasure to read the text of the submitted paper, considering the clear concept, chosen methodology, adequately processed and presented a large number of results, excellent writing style, and language, and, equally importantly, very nicely presented and discussed results, with guidelines for further work in this exciting area of research. Kudos to the authors.

I have no specific complaints except for those that can be classified as technical imperfections.

Please put all Latin expressions in italics (e.g., via), standardize the writing of the expression in vitro (there is no need to have a hyphen in between, i.e., to write in-vitro), correct minor typographical errors (e.g., an extra comma in a sentence on line 132, the mark for micro on line 753, as well as the abbreviation for the catalog number in the same place in the text (e.g., put CAT No.), or the space between the number and the measure mark: one example is on line 778).

We cleaned the text, in this respect.

 I also believe that reference number 26 is not entirely correctly cited (is it this article?: https://www.cadth.ca/sites/default/files/cdr/clinical/sr0603-tegsedi-clinical-review-report .pdf).

YES. However, we removed it along with the sentence “The shortcomings of the approved therapies involving tafamidis, inotersen, patisaran and diflusinal [26] do call for additional ATTR therapies” to avoid any confusion.

We thank the referee for the supportive comments and the kudos.

Reviewer 4 Report

Comments and Suggestions for Authors

The article is well written and has clear structure.

The English is fine, I found no issues, which could hinder accessibility for readers. The research is full and self-sufficient.

The only wish I can express is to employ already established TTR-related or new appropriate in vivo models/clinical samples to this research in the future, maybe in collaboration with a strong molbiol/clinicians team. 

Despite the rare incidence of ATTR, this field looks important and application of in vivo models can reveal some interplay with other amyloidosis-related diseases. 

Author Response

The article is well written and has clear structure.

The English is fine, I found no issues, which could hinder accessibility for readers. The research is full and self-sufficient.

The only wish I can express is to employ already established TTR-related or new appropriate in vivo models/clinical samples to this research in the future, maybe in collaboration with a strong molbiol/clinicians team. 

Despite the rare incidence of ATTR, this field looks important and application of in vivo models can reveal some interplay with other amyloidosis-related diseases. 

We thank the referee for the supportive comments. We would be deligthed to be contacted or to be put in touch with/by a molbiol/clinicians team to carry on research using in vivo models/clinical samples.